# Recent Advances in Water-Splitting Electrocatalysts Based on Electrodeposition

**DOI:** 10.3390/ma16083044

**Published:** 2023-04-12

**Authors:** Yujin Kim, Sang Eon Jun, Goeun Lee, Seunghoon Nam, Ho Won Jang, Sun Hwa Park, Ki Chang Kwon

**Affiliations:** 1Smart Device Team, Interdisciplinary Materials Measurement Institute, Korea Research Institute of Standards and Science (KRISS), Daejeon 34133, Republic of Korea; 2Department of Materials Science and Engineering, Andong National University, Andong 36729, Republic of Korea; 3Department of Materials Science and Engineering, Research Institute of Advanced Materials, Seoul National University, Seoul 08826, Republic of Korea

**Keywords:** water splitting, electrodeposition, electrocatalysts, layered double hydroxides, single-atom catalysts

## Abstract

Green hydrogen is being considered as a next-generation sustainable energy source. It is created electrochemically by water splitting with renewable electricity such as wind, geothermal, solar, and hydropower. The development of electrocatalysts is crucial for the practical production of green hydrogen in order to achieve highly efficient water-splitting systems. Due to its advantages of being environmentally friendly, economically advantageous, and scalable for practical application, electrodeposition is widely used to prepare electrocatalysts. There are still some restrictions on the ability to create highly effective electrocatalysts using electrodeposition owing to the extremely complicated variables required to deposit uniform and large numbers of catalytic active sites. In this review article, we focus on recent advancements in the field of electrodeposition for water splitting, as well as a number of strategies to address current issues. The highly catalytic electrodeposited catalyst systems, including nanostructured layered double hydroxides (LDHs), single-atom catalysts (SACs), high-entropy alloys (HEAs), and core-shell structures, are intensively discussed. Lastly, we offer solutions to current problems and the potential of electrodeposition in upcoming water-splitting electrocatalysts.

## 1. Introduction

Environmental pollution problems are increasing due to the unsustainable exploitation and use of fossil fuel [1,2,3,4]. The world is turning toward sustainable energy, with renewable energy being the most in demand [5,6,7,8,9]. Among renewable energy sources, hydrogen energy is rapidly gaining popularity due to its high calorific value and the fact that it returns to water after burning, making it a clean energy source [10,11,12,13,14,15,16,17]. However, several bottlenecks need to be addressed before hydrogen can be utilized as a sustainable fuel [1,18,19,20,21]. Hydrogen cannot be used in its natural state and requires fuel cell technology, which still needs further development [16,17,22,23,24,25,26,27]. Additionally, the commonly used method to produce hydrogen, steam reforming of fossil fuels, can emit large amounts of carbon dioxide. To overcome these challenges, the development of carbon dioxide capture, storage, reduction, and utilization technologies is necessary to complete the carbon neutralization process [28,29]. These technologies will not only reduce carbon dioxide emissions but also improve the efficiency of hydrogen production and storage [30]. As such, they play a critical role in the transition toward a sustainable energy future [31,32].

There are ongoing research and development efforts aimed at improving the efficiency of hydrogen production through water electrolysis, as well as reducing the cost of the production process. The hydrogen evolution reaction (HER) and oxygen evolution reaction (OER) are two important surface reactions in water electrolysis. The mechanism of these reactions depends on the pH level of the electrolyte. The reactions of hydrogen and oxygen evolution in an acidic medium follow the below mechanisms.
H^+^ + e^−^ → H_ad_,
H^+^ + e^−^ + H_ad_ = H_2_,
2H_ad_ → H_2_,
H_2_O (l) + ^∗^ → ^∗^OH + H^+^ + e^−^,
^∗^OH → ^∗^O + H^+^ + e^−^,
H_2_O (l) + ^∗^O → ^∗^OOH + H^+^ + e^−^,
^∗^OOH → ^∗^ + O_2_ (g) + H^+^ + e^−^,
where ^∗^ represents the active sites of the catalyst, (g) refers to the gas phase, and ^∗^OH, ^∗^O, and ^∗^OOH represent the species adsorbed on the active site [33]. The HER mechanism involves the reduction of hydrogen ions (H^+^) to form hydrogen gas (H_2_) at the cathode. The OER mechanism, on the other hand, involves the oxidation of water (H_2_O) to form oxygen gas (O_2_) at the anode. The reactions of hydrogen and oxygen evolution in an alkaline medium follow the below mechanisms.
H_2_O + e^−^ → OH^−^ + H_ad_,(1)
H_2_O + e^−^ + H_ad_ → OH^−^ + H_2_,(2)
^∗^ + OH^−^ → ^∗^OH + e^−^,(3)
^∗^OH + OH^−^ → ^∗^O + H_2_O (l) + e^−^,(4)
^∗^O + OH^−^ → ^∗^OOH + e^−^,(5)
^∗^OOH + OH^−^ → ^∗^ + O_2_ (g) + H_2_O (l) + e^−^.(6)

The HER mechanism involves the reduction of water (H_2_O) to form hydroxide ions (OH^−^) and hydrogen gas (H_2_) at the cathode. The OER mechanism involves the oxidation of hydroxide ions (OH^−^) to form oxygen gas (O_2_) and water (H_2_O) at the anode.

By combining water electrolysis and renewable energy, it is possible to produce “green hydrogen” without carbon emission [34,35]. The further development of new materials and advanced technologies, such as high-temperature electrolysis and photovoltaic electrolysis systems, has the potential to significantly increase efficiency and reduce the cost of hydrogen production [36,37,38]. The commercialization of these technologies would further enhance the competitiveness of hydrogen energy compared to other renewable energy sources and make it a more attractive option for large-scale energy production [39,40,41,42,43,44,45]. The transition to a hydrogen-based energy system is a complex and challenging process, but it holds great potential for mitigating the environmental pollution problems caused by the use of fossil fuels [10,41,42,43]. While there are still some technical and economic hurdles to overcome, the continued development and deployment of new technologies and innovative solutions are likely to soon lead to the wider adoption of hydrogen energy as a sustainable fuel [46,47,48,49,50,51,52,53].

Additionally, the choice of electrolyte plays a crucial role in the efficiency of electrolysis, as well as the stability of the electrolyte in the long term. Proton exchange membrane (PEM) electrolysis uses a thin polymer membrane as the electrolyte and is most suitable for high-pressure, high-volume hydrogen production, but it shows relatively low stability due to the harsh acidic operation. Alkaline-based water electrolysis uses a high-pH (basic) solution as the electrolyte, which is typically made of potassium hydroxide. This method is widely used in industrial applications due to its high efficiency and reliability [54,55,56]. In particular, anion exchange membrane (AEM) water electrolysis is considered more environmentally friendly than other methods, originating from its lower cell voltage drop and increased electrical conductivity. Furthermore, the AEM also operates under low-temperature and low-pressure conditions, reducing the need for cooling and compression systems, which results in a low carbon footprint. Despite these advantages, AEM technology is still in its early stages of development and requires further research for commercialization [57,58].

The catalyst plays a crucial role in reducing the overpotential and increasing the efficiency of the water electrolysis process. Currently, noble metal catalysts, such as Pt-based catalysts for hydrogen generation reactions and Ir-, Ru-based catalysts for oxygen generation reactions, are the best-performing materials [59,60,61]. These metals have unique electronic structures that allow for rapid oxidation-reduction reactions. However, the high cost of these metals has limited their practical use in large-scale hydrogen production [62,63,64]. To overcome this limitation, researchers have been actively seeking alternative catalysts that are more cost-effective, earth-abundant, and environmentally friendly [65,66,67,68,69]. Several transition metal compounds and alloys, such as iron, nickel, and cobalt, have shown promising results as water electrolysis catalysts [70,71,72,73,74,75,76,77,78,79,80,81,82]. In addition, the development of composite catalysts and hybrid catalysts has opened up new possibilities for enhancing the efficiency and stability of water electrolysis [83,84,85,86,87,88,89,90,91]. There are various methods of synthesizing those catalysts, including chemical vapor deposition, hydrothermal, corrosion, and electrodeposition [92,93]. Compared to other methods, electrodeposition offers a range of advantages. Electrodeposition allows for precise control over the shape and size of the materials, making it ideal for obtaining catalysts with specific dimensions and properties. Moreover, it can be utilized to synthesize a variety of materials including metals, oxides, sulfides, and phosphides. Furthermore, electrodeposition is environmentally friendly and economical, as it allows for the reuse of the electrolyte and results in stable catalysts without the need for additional treatments [94,95,96]. Interestingly, the unique structures and morphology such as nanowires, nanotubes, nanoparticles, and nanosheets achieved by electrodeposition enable tailoring the surface area, electronic properties, and surface reactivity, all of which affect the performance of catalysts. In addition, the atomic structure of electrodeposited catalysts can be elaborately modified to boost the activity of catalytic sites. To further elaborate, electrodeposition is being explored as a method to increase production for industrial applications. Recently, a research paper reported the successful electrodeposition of an electrocatalyst onto a substrate that was 136 cm^2^ in size [97]. This is a noticeable figure even in previous research; on this basis, nanostructure design is actively used in the research field of electrocatalysts to obtain excellent performance.

Moreover, the use of electrodeposition for water-splitting catalysts can be a promising candidate for the development of cost-effective and efficient water-splitting catalysts for industrial-scale hydrogen products [98,99]. Recently, layered double hydroxides, single-atom catalysts, and high-entropy alloys have attracted significant attention in the field of electrocatalysis due to their extraordinary catalytic activity derived from a unique atomic structure [100]. Using various techniques in electrochemical deposition such as galvanostatic, potentiostatic, pulsed, and cyclic voltammetry modes, they can be uniformly synthesized, and their catalytic activity can be boosted. Further research is needed to optimize these alternative catalysts and to make water electrolysis a more practical and economically viable method for hydrogen production. In this review, we highlight the recent development and several strategies in the field of electrodeposition for water splitting to overcome the current challenges.

## 2. Mechanism and Mode of Electrodeposition

### 2.1. Various Modes in Electrodeposition

For electrochemical energy technologies, electrodeposition is a promising way to synthesize nanostructured electrocatalysts. The method can grow components in a short time with facile control of a uniform nanoscale morphology by applying potential or current [101,102]. Furthermore, the chemical stability between deposited catalysts and substrate can be realized due to electrochemical chemisorption, resulting in an excellent water-splitting performance with high stability [101]. Herein, we introduce the electricity input approach of electrodeposition and the strategies to form nanostructured catalysts.

#### 2.1.1. Galvanostatic and Potentiostatic Modes

For electrodeposition, electricity is applied in the form of current density or potential. Galvanostatic mode is performed by controlling the number of charges inputted with a constant current density without reference to an electrode [103]. In this mode, the current density can be directly controlled to determine the number of nucleation and deposited amounts [104]. In contrast, the potentiostatic mode is a method of depositing materials using a constant input potential. Applying a constant potential typically results in the deposition of a single, pure phase [103]. Since the potential difference between the counter electrode and working electrode is an important factor, potentiostatic mode is usually performed in a three-electrode system with a reference electrode. The applied potential is determined by the redox potential of the element to be deposited. When the applied potential is negative relative to the equilibrium potential, it is referred to as underpotential deposition. The deposition rate, amount, shape, and composition of the catalysts can be determined by the applied potential or current density; hence, it must be carefully controlled. Furthermore, resistance in the electrolyte and the distance between electrodes can also play important roles when controlling the potential or current density.

#### 2.1.2. Pulsed Mode

The morphology of a nanostructured electrocatalyst can be easily tuned by controlling the applied electricity. The pulse setting can be modulated using specific frequencies with different input potentials or current densities. For instance, by applying a pulse group including cathodic and anodic current signals with specific frequency to substrates, elements can be grown and etched away when using a negative redox potential metal source. Nucleation grows anisotropically and vertically from the substrate, making it possible to assemble a nanoarray structure in a single step through electrodeposition. In this way, active sites can be increased, and catalytic performance can be enhanced by controlling the growth behavior of crystal nucleation using pulsed mode. Additionally, the pulsed mode, widely used in catalyst synthesis, can reduce diffusion resistance and control the microstructure and multicomponent composition [104].

#### 2.1.3. Cyclic Voltammetry (CV) Mode

Recently, CV (cyclic voltammetry) mode has been studied for depositing nanostructured and atomic-scale catalysts. In CV mode, a dynamic potential is applied, and the corresponding current density is measured [104,105]. Generally, when a low current density is measured, the electrolyte resistance increases due to a small amount of charge, making it difficult to deposit a uniform coating and resulting in a slow deposition rate [106,107]. Despite these challenges, some studies have attempted to electrochemically deposit various components, including nanostructures and heteroatom doping, as well as atomic-scale elements [108].

### 2.2. Various Modes in Electrodeposition

As discussed in the previous section, electrodeposition is an effective way of synthesizing electrocatalysts due to the ease in controlling the morphology, deposition amount, nanostructures, and atomic deposition. Liu et al. used linear sweep voltammetry (LSV) for the electrochemical activation and modification of WO_3_@Cu foam (WO_3_@CF) with Pt, as shown in Figure 1a [109]. They synthesized WO_3_ nanosheet arrays on CF through a solvothermal method and modified the heterostructure catalysts using LSV mode and a Pt counter-electrode. The Pt clusters preferentially deposited on dissolved Cu sites rather than WO_3_ due to the small lattice mismatch between Pt and Cu in the acidic electrolyte (0.5 M H_2_SO_4_). They observed excellent Pt mass activity in optimized Pt-WO_3_@CF catalysts for the hydrogen evolution reaction (HER). High-entropy alloys are considered promising as electrocatalysts due to their cost-effectiveness and the synergistic effects of each element, as demonstrated by Chang et al. [110]. In their study, they used metal precursors of Fe, Co, Ni, Mn, and W in an electrolyte and performed electrodeposition using pulsed current mode for the synthesis of a bifunctional electrocatalyst for water splitting. FeCoNiMnW high-entropy alloys with heterostructure were uniformly and well dispersed on a carbon paper substrate. Through this simple pulsed electrodeposition method, the high-entropy alloy showed enhanced catalytic activity. Another strategy using single-atom catalysts has been the focus of intense study due to their high mass catalytic activity and low mass loading of noble metal. Furthermore, noble or transition metals at the atomic scale provide numerous active sites and great surface area compared to even nanoparticles. Zhang et al. reported the synthesis of Ir single-atom catalysts (SACs) on a nanosheet substrate using electrochemical deposition as a universal route [111]. They used linear sweep voltammetry (LSV) to deposit the precious metal on the prepared substrate both cathodically and anodically. The electrolyte contained a minimum amount of noble metal precursor, which was efficiently utilized for water splitting. Electrodeposition is an effective method to design nanostructured catalysts for electrochemical water splitting. In the next sections, we investigate previous strategies for enhancing catalytic activity using various electrodeposition modes and introduce different nanostructures.

## 3. Electrocatalysts Prepared by Electrodeposition for Water Splitting

### 3.1. Layered Double Hydroxides (LDHs)

Recently, the study of layered double hydroxides (LDHs) as OER catalysts has gained popularity due to their porous morphology and favorable adsorption energy at the surface, which is derived from their unique atomic structure [112,113,114,115,116,117,118,119,120,121,122,123,124]. LDHs are a class of anionic clay with a layered structure defined as the atomic formula [M_1−x_^2^ + M_x_^3^ + (OH)_2_](A^n−^)_x/n_·mH_2_O, where M^2+^ and M^3+^ are divalent and trivalent metal cations, and A^n−^ represents a charge-balancing anion. Li et al. [125] synthesized a heterostructure of CoNiP and NiFe LDHs via a two-step electrodeposition method, which is shown in Figure 2a. The CoNiP nanoparticles with around 100 nm were deposited at a constant potential of −1.6 V (vs. Ag/AgCl) for 800 s using a solution containing NiSO_4_, CoSO_4_, NaH_2_PO_2_, NaCl, H_3_BO_3_, and sodium citrate, with boric acid serving as a buffering agent and sodium citrate as complexes. The NiFe LDHs were deposited at a constant potential of −1.0 V (vs. SCE) for 120 s using a solution containing Ni(NO_3_)_2_·6H_2_O and Fe(SO_4_)_2_·7H_2_O. Figure 2b shows the vertically aligned NiFe LDH nanosheets grown on the surface of electrodeposited CoNiP nanoparticles stacked on the nickel foam. In this image, the agglomeration of the LDH nanosheets can be observed. The LDH nanosheets tend to agglomerate due to strong interlayer van der Waals forces, resulting in decreased surface area and performance. Therefore, it is required to obtain uniformly dispersed LDH nanosheets using surfactants and pH control agents [126]. High-resolution transmission electron microscope (HR-TEM) images showed that the amorphous NiFe LDH nanosheets adequately covered the low-crystalline CoNiP nanoparticles, and only a small microcrystalline region can be found, as displayed in Figure 2c. Additionally, the uniform distribution and coexistence of Ni, Fe, O, Co, and P elements can be observed in the TEM elemental mapping images. The HER and OER performance of the as-synthesized CoNiP@NiFe LDH was evaluated by electrochemical measurements, as shown in Figure 2d,e. The CoNiP@NiFe LDHs only required a low overpotential of 68 mV to achieve a current density of 10 mA/cm^2^, comparable to that of CoNiP (80 mV) and NiFe LDH (283 mV). Furthermore, the Tafel slope, determined from the kinetics of HER, of CoNiP@NiFe LDH was calculated to be 32 mV/dec, also comparable to that of CoNiP (41 mV/dec) and NiFe LDH (142 mV/dec), indicating that the CoNiP@NiFe LDH followed fast HER kinetics, i.e., a Volmer–Tafel mechanism. According to the electrochemical impedance spectra (EIS), a significantly low charge transport resistance could be found in CoNiP@NiFe LDH (0.15 Ω), confirming the favorable reaction kinetics and better charge transfer efficiency of CoNiP@NiFe LDH. For the OER, the CoNiP@NiFe LDH also exhibited remarkable OER performance with a low overpotential of 255 mV at 50 mA/cm^2^. The overpotential of as-prepared CoNiP and NiFe LDH was 370 mV and 280 mV at 50 mA/cm^2^, respectively, indicating that the NiFe LDH played a key role in the catalytic activity of the OER process and synergistic effect between NiFe LDH and CoNiP in OER process. The prepared CoNiP@NiFe LDH catalysts showed excellent stability in alkaline medium over 20 h with an increased overpotential of 7 mV for HER and over 12 h with a slightly increased overpotential of 1.8 mV for OER. A DFT calculation was carried out to identify the synergistic effect of the heterostructures. In the case of HER, the CoNiP@NiFe LDHs (ΔH^*^ = −0.62 eV) had a considerably lower hydrogen adsorption Gibbs free energy than the CoNiP and NiFe LDHs (ΔH^*^ = −0.75 eV for CoNiP and ΔH^*^ = −0.87 eV for NiFe LDHs), as shown in Figure 2f. For OER, in NiFe LDHs, as shown in Figure 2g, the rate-determining step was O^*^ OOH^*^ (Δ_GOOH*_ = 2.79 eV). However, that of CoNiP@NiFe was the formation of oxygen (ΔGO_2_ = 2.01 eV), indicating that an easier adsorption of OOH^*^ intermediates could be found in CoNiP@NiFe LDHs, originating from the electronic coupling between CoNiP and NiFe LDHs. This result shows that the heterostructure required a lower overpotential and had higher catalytic activity in OER reactions.

Yamauchi and colleagues conducted a study on NiFe and W-doped NiFe LDHs that were obtained by combining electrodeposition and chemical corrosion [127]. To create the NiFeW_x_-LDHs, a cathodic current was applied to a solution of Ni(NO_3_)_2_·6H_2_O and Fe(NO_3_)_3_·9H_2_O for 1 h. The resulting NiFe LDHs were then immersed in a WCl_6_ solution for 3 h to introduce oxygen vacancies, which converted the NiFe LDHs into NiFeW LDHs, as depicted in Figure 3a. Scanning electron microscope (SEM) measurements showed changes in surface morphology, including the presence of nanopores, which could increase the contact area between the electrolytes and catalysts, resulting in faster desorption of OER products. A TEM image showed that the NiFeW_3_-LDHs had a nanosheet structure consisting of polycrystalline regions with good lattice fringes and amorphous phases, which were confirmed by SAED patterns, as displayed in Figure 3b. The authors studied the electronic interactions in NiFeW_3_-LDHs by analyzing the valence electron structures of the metal ions. They found that partial electron transfer from Ni^2+^ to Fe^3+^ occurred due to the π-donation from the O^2−^ to d-orbitals of Fe^3+^ and electron–electron repulsion between O^2−^ and d-orbitals of Ni^2+^. Then, W^6+^ obtained electrons from the electron-rich Fe^3+^, leading to enhanced delocalization among Ni, Fe, and W cations [122]. The modified charge configuration optimized the bonding strength of the cations, as Ni^2+^ and Fe^3+^ were weakly bonded to oxygen-adsorbed species, while W^6+^ was strongly bonded [128]. In Figure 3c, the LSV curves of NiFe LDHs and NiFeW_x_-LDHs are provided to investigate the electrochemical catalytic performance with 90% iR correction. The overpotential of NiFeW_3_-LDHs was 256 mV to achieve a current density of 100 mA/cm^2^, which was smaller than that of NiFe LDHs, NiFeW_1_-LDHs, and NiFeW_5_-LDHs. For the Tafel slope, as illustrated in Figure 3d, a small value of 36.44 mV/dec could be calculated for NiFeW_3_ LDHs, indicating a good high-current electrocatalytic performance. The lower Tafel slope value indicated faster reaction kinetics, which could lead to higher efficiency and improved performance of water-splitting devices such as electrolyzers or photoelectrochemical cells [129]. These outperformances mainly originated from the porous and rough surface of NiFeW_x_-LDH, which increased the contact area between electrolytes and catalysts. In Figure 3e, the LSV curves of the NiFeW_3_-LDHs obtained at various temperatures were presented to calculate the activation energy (E_a_). The activation energy of NiFeW_3_-LDHs was 24.66 kJ/mol, which was smaller than that of NiFe LDHs, implying that W doping significantly reduced the energy barrier of the OER. From the chronopotentiometry and cycle test for 120 h at 10 mA/cm^2^ and 5000 cycles, the NiFeW_3_-LDH retained its initial catalytic performance without significant changes. The nanopore structure could be maintained even after 5000 CV cycles and long-term operation. This concept provides a route for utilizing the oxygen vacancies via foreign element doping to improve the performance of OER electrocatalysts.

### 3.2. Single-Atom Catalysts

Single-atom catalysts (SACs), which are catalysts made up of isolated metal atoms dispersed on a support material, have received a lot of attention due to their unique properties and high catalytic activity. Some of the advantages of SACs include high activity due to their high surface area-to-volume ratios, precise control over reaction selectivity, and durability in harsh industrial conditions [130,131]. However, the synthesis of SACs can be challenging, their scalability is limited, and they can be more expensive than traditional catalysts [132,133]. To overcome these limitations, researchers are exploring new synthetic methods to produce SACs at a lower cost and on a larger scale, investigating new support materials and alternative metal species, and using advanced characterization techniques to better understand the structure–activity relationships of SACs.

Yin et al. reported an efficient strategy to synthesize Ir single-atom catalysts (Ir-SACs) at NiCo_2_O_4_ (Ir-NiCo_2_O_4_) nanosheets (NSs) on carbon cloth (CC) as a substrate using a co-electrodeposition method and oxidation process, as shown in Figure 4a (structural characterization) [134]. The precursor solutions consisted of nickel(II) nitrate, cobalt(II) nitrate, and chloroiridic acid, and a constant potential of −1.0 V was applied to the working electrode for 15 min. After the electrodeposition, they conducted annealing in air conditioning for Ir-SA coupling with oxygen vacancy. The Ir atoms could locate the oxygen vacancy site, especially near the Co sites, resulting in boosting the OER performance due to the activation of electron transfer. The tight Ir-O bonding played a role in inhibiting deactivation of the catalytic active site due to the supersaturation of O and H at the Co sites. The well-orientated spinel structure of NiCo_2_O_4_ [111] unit cells and the extensively dispersed Ir SAs, marked by a yellow circle, were observed in high-angle annular dark-field scanning transmission electron microscope (HAADF-STEM) images (Figure 4b,c, respectively). To elucidate the effect of Ir-SACs on NiCo_2_O_4_ NSs, electrochemical analysis was performed by measuring LSV, turnover frequency (TOF), EIS, and chronoamperometry in acidic medium (0.5 M H_2_SO_4_). Ir-NiCo_2_O_4_ NS exhibited the lowest overpotential of 240 mV vs. RHE at 10 mA/cm^2^ and the best Tafel slope of 60 mV/dec among the prepared catalysts, as shown in Figure 4d. To compare the OER performance using Ir SAs on oxide-based nanosheets, they utilized various transition metal oxide nanosheets as a substrate. Among the various oxide nanosheets, as shown in Figure 4e, the Ir-NiCo_2_O_4_ NSs showed the best catalytic performance due to the enhanced electron transfer properties and the enhanced content of oxygen in the Ir-NiCo_2_O_4_ crystal structure. In addition to experimental results, they compared recently reported OER catalysts, as displayed in Figure 4f. The synthesized Ir-NiCo_2_O_4_ NSs showed relatively good OER performance and enhanced stability in acidic medium over 70 h, maintaining their initial properties without any degradation of composition, morphology, or electronic structure in further investigation after OER measurement. The enhanced stability even in acidic medium originated from the increased oxygen content from the surface reconstruction of Ir-O_x_. This proposed strategy suggests that the improvement of catalytic performance in energy conversion reactions can be achieved by a simple preparation method to prepare SACs on transition metal oxide-based electrocatalysts.

Zhang et al. reported a facile method to synthesize Ir SAs on Co_0.8_Fe_0.2_Se_2_ for overall water splitting using simple linear sweep voltammetry (LSV) [111]. Cathodically and anodically deposited Ir SAs for HER and OER, respectively, were conducted by LSV in KOH electrolyte containing 100 μM IrCl_3_. For the electrodeposition, the as-prepared MnO_2_, MnS_2_, Co_0.8_Fe_0.2_Se_2_, and nitrogen-doped carbon (N-C) substrate was used as the working electrode and the Ag/AgCl electrode was used as the reference electrode. For the counter-electrode, a graphite rod was used. Each substrate was soaked into an electrolyte bath, and a potential from 0.10 V to −0.40 V was applied for cathodic deposition with 10 scanning cycles. In the case of anodic deposition, the potential from 1.1 V to 1.8 V was applied for three scanning cycles using LSV mode. Then, Ir cation (IrCl^3+^) and Ir anion (Ir(OH)^6−^) complexes could be deposited by scanning cycles; as a result, Ir-SAs were deposited on the substrate as illustrated in Figure 5a,b. For the deposition of Ir SAs, the Ir^4+^ precursor was reduced under the negative electric field and cooperated with OH^−^ in the KOH solution under a positive electric field. Figure 5c,d demonstrates Ir mass loading as a function of 50–300 μM Ir precursor concentration in the 1 M KOH electrolyte for cathodic and anodic deposition scanning cycles. From the HAADF-STEM measurement, the uniformly deposited Ir-SAs, indicated by yellow circles, were observed in the synthesized C- and A-Ir_1_/Co_0.8_Fe_0.2_Se_2_ catalysts, as shown in Figure 5e,f. The dispersed metal with atomic scales could be anchored at defective or strong metal support interactions. It is important that the mass loading is controlled below the supersaturated level unless the mass loading above the supersaturated level leads to nuclearization into a cluster. They conducted electrochemical overall water-splitting measurements. The overpotential of the cathodically deposited Ir SAs on Co_0.8_Fe_0.2_Se_2_@Ni foam catalyst (C-Ir_1_/Co_0.8_Fe_0.2_Se_2_ catalyst) was 8 mV at 10 mA/cm^2^ for hydrogen evolution, and the overpotential of the anodically deposited Ir SAs on Co_0.8_Fe_0.2_Se_2_@Ni foam catalyst (A-Ir_1_/Co_0.8_Fe_0.2_Se_2_) was 230 mV at 10 mA/cm^2^ for oxygen evolution, as displayed in Figure 5g,h. According to the experimental results, Ir_1_/Co_0.8_Fe_0.2_Se_2_ catalyst exhibited outstanding catalytic activity for overall water splitting in alkaline medium. They also discussed that the Co_0.8_Fe_0.2_Se_2_ substrate exhibited excellent catalytic activity for overall water splitting when anchored with Ir single atoms due to its excellent charge transfer and strong Ir metal single atom–Co_0.8_Fe_0.2_Se_2_ support interaction. Furthermore, the synthesized Ir SAs on Co_0.8_Fe_0.2_Se_2_@Ni foam catalyst showed long-term stability over 100 h in alkaline medium (1 M KOH), ascribed to the distinct deposition method.

Wang et al. reported Ru single-atom catalysts (Ru SACs, by electrodeposition) incorporated with MoS_2_ nanosheets (NSs, by hydrothermal) on carbon cloth [135]. Although Pt shows the best catalytic performance for the HER because of its high adsorption efficiency of H^+^ originating from its unique electronic structure, it is too expensive to commercialize. Therefore, recently, atomic-scale noble metal catalysts have been intensively studied to reduce the usage of noble metal catalysts or to replace noble metal catalysts with non-noble catalysts. Ru is relatively inexpensive compared to Pt and has a similar electronic structure, indicating that it is not only suitable for replacing Pt but also widely used as SACs in HER [136,137]. First, they synthesized MoS_2_ nanosheets on a carbon cloth (MoS_2_ NSs/CC) array using a hydrothermal method with sodium molybdate and thiourea precursor dissolved in deionized water. After the hydrothermal method, the electrodeposition of Ru SACs was conducted using MoS_2_ NSs/CC as a deposition substrate, as displayed in Figure 6a. The deposition potential from −0.5 V to 0.4 V with a sweep rate of 20 mV/s was applied to MoS_2_ NSs/CC substrate in RuCl_3_ and H_2_SO_4_ electrolyte for 20 cycles; then, the Ru precursor formed SAs on MoS_2_ NSs/CC. Figure 6b shows the morphology of electrodeposited MoS_2_ NSs/CC substrate with vertically and densely aligned nanosheet structures. The MoS_2_ (100) and (002) planes were observed in HR-TEM images with d-spacings of 0.274 and 0.65 nm, and the electrochemically deposited Ru atoms were dispersed at the atomic scale in the synthesized Ru-MoS_2_/CC [137,138], marked by black circles in Figure 6c. They measured electrochemical HER performance in both alkaline and acidic electrolytes to reveal the overall water-splitting catalytic activity of the Ru-MoS_2_/CC electrode. In alkaline medium (1 M KOH), the overpotential of Ru-MoS_2_/CC was 41 and 11371 mV at 10 and 100 mA/cm^2^, respectively, and a Tafel slope of 114 mV/dec could be obtained, as shown in Figure 6d,e. Furthermore, the efficient HER catalytic activity in acidic (0.5 M H_2_SO_4_) and neutral (1 M PBS) media was confirmed with a low overpotential of 61 mV and 114 mV at 10 mA/cm^2^, respectively. They conducted the chronopotentiometry of HER using Ru-MoS_2_/CC and Pt/C/CC electrocatalysts at 10 mA/cm^2^ to test the electrochemical stability, as depicted in Figure 6f. The Ru-MoS_2_/CC electrode retained its initial catalytic performance after 20 h, whereas that of Pt/C/CC decreased gradually. According to further XPS, SEM, and HR-TEM analysis, there was no significant structural degradation and dissolution of Ru SAs. The results strongly indicate that these approaches to electrodeposit SAs could be promising to improve the catalytic HER performance of Ru-MoS_2_ electrocatalyst.

### 3.3. High-Entropy Alloys

Metal alloying is one of the promising strategies for synthesizing water-splitting catalysts, due to their synergistic effect on each component [139,140,141,142,143,144,145]. In particular, high-entropy alloys (HEAs) represent a fascinating method for increasing catalytic activity [146]. A strong synergistic effect is gained when the alloy component is widely contacted and closely interacted, resulting in an HEA with four or more components, which is a suitable candidate for water-splitting catalysts. In a recent report, Chang et al. studied a facile strategy to synthesize HEA-FeCoNiMnW (H-FeCoNiMnW) and its application to water splitting in an acidic medium [110]. They also demonstrated overall water splitting using HEA-FeCoNiMnW electrocatalysts as both the cathode and the anode.

They used carbon paper (CP) as a substrate for improving electrical conductivity and selected metal precursors with high catalytic performance to adsorb and desorb water-splitting intermediates. FeCl_3_, NiCl_2_⋅6H_2_O, MnCl_2_⋅4H_2_O, CoCl_2_⋅6H_2_O, Na_2_WO_4_⋅2H_2_O, and C_6_H_5_Na_3_O_7_∙2H_2_O were used to compose the electrolyte, and the pH value was set to 9.8 using NH_4_OH. The electrodeposition was conducted in pulse mode (current density of 8 A/cm^2^; 20% duty for 3000 cycles) with a bath temperature of 53 °C, as illustrated in Figure 7a. From the HR-TEM image in Figure 7b, an interlayer distance of 0.212 nm could be observed, consistent with the d-spacing of the face-centered cubic (111) plane of HEA-FeCoNiMnW. From the measurement of ICP-OES, the atomic ratio of all five elements could be calculated as 18.4%, 18.1%, 26.4%, 13.4%, and 23.7% for Fe, Co, Ni, Mn, and W, respectively. The broad characteristic (111) diffraction peak was found in XRD spectra, originating from the small grain size and large lattice strains of HEA crystals [147]. From Vegard’s rule, the lattice constant should be 0.369 nm, which is consistent with the calculated values of 0.367 nm and 0.36 nm from the HR-TEM image and XRD pattern, respectively. To investigate the catalytic performance of HEA in water splitting, electrochemical characterization was conducted in 0.5 M H_2_SO_4_, and H-FeCoNiMnW was compared to other medium-entropy alloy catalysts (M-FeCoNiMn and M-FeCoNiW), Pt/C for HER, and IrO_2_ for OER (see Figure 7e,f). H-FeCoNiMnW showed a low overpotential of 15, 73, and 165 mV at 10, 100, and 500 mA/cm^2^, respectively. For the MEA, a slightly increased overpotential could be found in M-FeCoNiMn (193, 307, and 416 mV for η_10_, η_100_, and η_500_) and M-FeCoNiW (32, 138, and 226 mV for η_10_, η_100_, and η_500_), due to the different catalytic activity of each element. When W was included in the FeCoNi alloy, the catalytic performance could be enhanced compared to Mn. In the case of Mn, the insertion of Mn into FeCoNiW alloy could boost the HER catalytic activity. The OER catalytic performance was investigated using IrO_2_, H-FeCoNiMnW, M-FeCoNiMn, and M-FeCoNiW with overpotentials of 332, 512, 608, and 595 mV at 10 mA/cm^2^, respectively. Furthermore, the Tafel slope of H-FeCoNiMnW was calculated to be 145 mV/dec, showing faster kinetics compared to MEAs with 161 mV/dec and 172 mV/dec, indicating that the HEA electrode was more efficient in electrochemical water splitting. The incorporation of Mn and W elements into FeCoNi helped to improve the catalytic activity. The overall water-splitting performance was investigated using H-FeCoNiMnW electrodes as both the cathode and the anode. The cell voltage required to achieve a current density of 10 mA/cm^2^ was 1.76 V, as shown in Figure 7g. A 10% increment in cell voltage was found in the H-FeCoNiMnW||H-FeCoNiMnW coupled electrode after 6 days of operation. This study is the first demonstration of the synthesis of high-entropy alloys based on a volcano plot with nonprecious metal precursors applied to water splitting in acidic medium. It provides a novel strategy to prepare efficient electrocatalysts in practical water splitting.

Han et al. generated β-NiOOH in amorphous high-entropy electrocatalysts for OER [148]. They mixed electrolytes using NiCl_2_, FeCl_2_, CoCl_2_, MnCl_2_, AlCl_3_, LiClO_4_, and ethylenediamine (EDA) in DMSO as precursors. Binary NiFe was chosen as the starting material; then, Co, Mn, and Al were doped to examine the relationship between the composition and OER performance. The high-entropy NiFeCoMnAl oxide could be prepared by electrodeposition with a potential of −2.8 V (vs. SCE) for 30 min. To create the nanoscale porous structure, the electrodeposited sample was soaked in 1.0 M KOH solution for 1 h to etch the Al element, as shown in Figure 8a. The increase in active surface areas and the creation of a synergistic effect for OER activity could be expected. Since the electrodeposition provided a strong binding between the catalyst and substrate with improved surface area, the electrodeposited catalysts could show high OER performance and excellent stability. A dense and uniform covered catalyst layer could be found in electrodeposited NiFeCoMnAl oxides on the surface of carbon fibers, as shown in the SEM images (Figure 9b). After the etching of Al elements by KOH, the nanoporous NiFeCoMnAl oxide nanosheet structure was observed by TEM images, as shown in Figure 8c,d. An amorphous structure was confirmed by the lack of significant Bragg reflection of the fresh sample in XRD spectra or lack of clear lattice fringes with diffuse rings in SAED patterns from the TEM analysis. The EDS mapping showed that the elements (Ni, Fe, Co, Mn, and Al) were homogeneously dispersed in the catalyst, as illustrated in Figure 9e. There was no structural degradation and obvious atomic aggregation in NiFeCoMnAl oxides after chemical etching, consistent with the EDS element mapping images. The electrochemical measurements of NiFeCoMnAl were carried out with a three-electrode system and compared to NiFe, NiFeAl, NiFeCoAl, RuO_2_, and bare CP. In addition, the authors annealed NiFeCoMnAl in air at 400 °C and measured OER performance. As displayed in Figure 8f, the overpotential at 10 mA/cm^2^ before and after annealing NiFeCoMnAl was 190 mV and 220 mV in 1 M KOH, respectively. The fabricated NiFeCoMnAl showed a low value of the Tafel slope (47.62 mV/dec) and long-term stability of over 50 h at 10 mA/cm^2^ without significant degradation, as shown in Figure 8f. The turnover frequency data showed that the NiFeCoMnAl catalysts (0.526 mol O_2_/s at η = 350 mV) had a much higher O_2_ production capability compared to NiFeCo (0.170 mol O_2_/s at η = 350 mV) and NiFeMn (0.138 mol O_2_/s at η = 300 mV). Operando Raman measurements showed that the active sites in NiFeCoMnAl oxide were Ni sites, and the high-valence Mn played a role in supporting the formation of β-NiOOH in an electron-rich environment. The results of density functional theory (DFT) calculations suggested that the self-constructed β-NiOOH synthesized by inflicting potential could improve OER activity [149,150]. Amorphous NiFeCoMnAl could maintain 98.6% of its initial potential even after 50 h at a current density of 10 mA/cm^2^. The authors also conducted a cyclic LSV test over 1000 cycles, revealing no obvious changes in morphology and component distribution, which indicated that the fabricated NiFeCoMnAl was an excellent OER electrocatalyst in an alkaline environment. Several factors contribute to enhanced OER activity and high stability:The self-construction of β-NiOOH intermediates, which can act as catalytic active sites, resulting in the lowering of the overpotential for the OER.Enhanced electrical conductivity, which can facilitate the improved charge transport resistance.The increased number of active sites from abundant defects via dealloying of Al. The study provided insight into the relationship between catalytic activity and elements and the design of multicomponent transition metal-based high-entropy catalysts for OER.

### 3.4. Core–Shell Structure

The enhancement of catalytic activity and improved stability using a nonprecious metal-based catalyst is the ultimate goal to realize highly efficient overall water splitting [43]. To seek effective bifunctional catalysts for simultaneously catalyzing HER and OER, there have recently neem tremendous efforts in finding a new type of electrocatalyst. The interface-engineered core–shell architectures consisting of two or more active catalysts are among the recent research themes to overcome the limitations of current electrocatalysts [151,152]. Improved charge transfer and increased opportunities for tuning the adsorption–desorption energy in water-splitting intermediates are expected in nanointerface core–shell structured electrocatalysts.
Figure 9(**a**) SEM images and (**b**) HR-TEM image of CoFe@NiFe-200/NF architecture. (**c**) XRD patterns of CoFe-LDH, NiFe-LDH, and CoFe@NiFe-200. (**d**) TEM, (**e**) magnified TEM, and (**f**) elemental mapping images of CoFe@NiFe-200 architecture. (The symbol of red and black represents NiFe-LDH and CoFe-LDH, respectively). (**g**) Steady-state polarization curves of CoFe-LDH/NF, CoFe@NiFe-50/NF, CoFe@NiFe-100/NF, CoFe@NiFe-200/NF, NiFe-300/NF, NiFe-LDH/NF, and NF in 1.0 M KOH for HER and OER. Overpotential comparison of synthesized catalysts (**h**) for HER and (**i**) for OER at a current density of 10 mA/cm^2^. Reprinted (adapted) from [153], copyright (2023) Elsevier B.V.
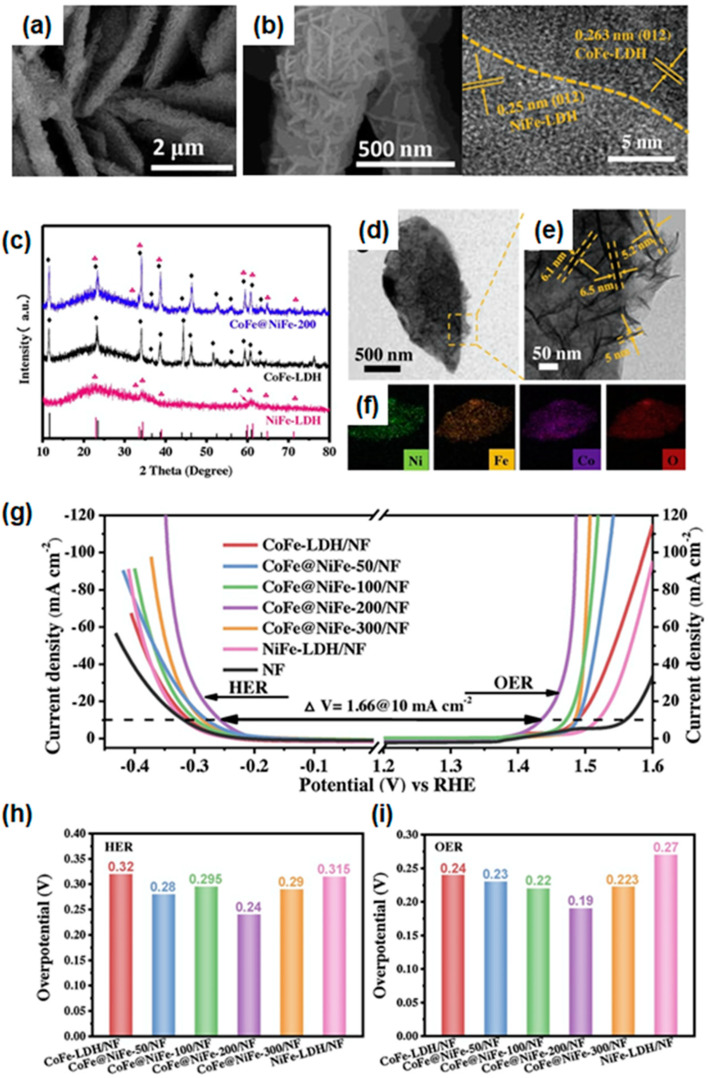



Yang et al. reported integrated 3D hierarchical CoFe-LDHs coupled with NiFe-LDH NS core–shell architectures as efficient overall water-splitting catalysts, as shown in Figure 9a [153]. To overcome the low electrical conductivity and water dissociation activity of CoFe-based LDHs, NiFe LDHs were adopted for a synergistic effect and to increase the active surface area. First, they synthesized the CoFe-LDHs on the Ni foam using the hydrothermal method and conducted electrodeposition of NiFe-LDHs by applying a potential of −1.0 V for 50 s, 100 s, 200 s, and 300 s. By controlling the deposition time, the loading amount of the NiFe-LDHs nanosheet shell could be modulated. A proper loading amount of NiFe-LDHs was important to expose the maximum interface or to optimize electronic interaction for the improvement of electrocatalytic activity in both HER and OER. The hierarchical interface and lattice fringes of the (012) plane of CoFe-LDH and NiFe-LDH could be found in TEM analysis, as displayed in Figure 9b. In the XRD patterns (Figure 9c), all the diffraction peaks in CoFe@NiFe LDHs could be assigned to the CoFe-LDHs and NiFe-LDHs without impurity phases. As shown in the HR-TEM images, the nanosheets with a thickness of about 6 nm were anchored to the smooth nanosheet of CoFe-LDH, forming core–shell architectures, as displayed in Figure 9d,e. The Co, Fe, and Ni elements were well distributed in the elemental mapping image (Figure 9f), indicating that the NiFe-LDHs were uniformly grown on the surface of CoFe-LHD, forming a core–shell structure.

The authors investigated the electrochemical catalytic performance of HER and OER, and examined the overall water-splitting performance by coupling their catalysts, as displayed in Figure 9g. In HER, the CoFe@NiFe-200 sample showed the highest catalytic HER performance with a low overpotential of 240 mV at 10 mA/cm^2^ and a Tafel slope of 88.88 mV/dec. The outstanding HER kinetics could be attributed to the synergistic effect between CoFe and NiFe LDHs, which was comparable to the previously reported catalysts. Furthermore, the CoFe@NiFe-200 sample exhibited remarkable catalytic performance in OER with a low overpotential of 190 mV at 10 mA/cm^2^ and a small Tafel slope of 45.71 mV/dec, originating from the low charge transport resistance and an enlarged number of active sites. The summarized water-splitting performance can be seen in Figure 9h,i. The enhanced stability without significant degradation was confirmed by conducting a cyclic test over 1000 cycles in both HER and OER. To investigate the overall water-splitting performance, they fabricated a two-electrode system by assembling CoFe@NiFe-200 catalysts for the cathode and anode. Outstanding performance with a low cell voltage of 1.59 V at 10 mA/cm^2^ was observed, and the initial current density was maintained after 24 h. This work paved the way to design highly efficient electrocatalysts using core–shell structures for bifunctional water splitting, as well as strategies for the synergistic effect of interfacing materials for improved catalytic activity.

Lee’s group demonstrated CuNi@Ni(ON) core–shell heterostructures uniformly dispersed on 3D porous CNTs-Gr for alkaline HER and OER using an electrodeposition method [154]. They conducted 8 CV cycles in the potential range of −0.5 to −1.0 V vs. Ag/AgCl with a scan rate of 5 mV/s in the solution of 1.0 mM Cu(Co_2_CH_3_)_2_ and 40 mM Ni(NO_3_)_2_ 6H_2_O, resulting in the formation of CuNi@Ni core–shell nanoparticles. Then, a partial nitridation treatment was carried out in a quenching furnace at 400 °C for 2 h under an atmosphere of Ar (100 sccm) and NH_3_ (50 sccm). The resultant CuNi@Ni(ON) heterostructures on CNTs-Gr exhibited dual functionality in promoting both HER and OER. This was attributed to alterations in the electronic structure of the surface, the type and number of electroactive sites, and charge conductivity. Moreover, the 3D porous and highly conductive CNTs-Gr substrate enhanced the stabilization of the active materials, facilitated hetero-charge transfer, and accelerated mass transfer. In OER, CuNi@Ni(ON)/CNTs-Gr showed the highest performance with an overpotential of 410 mV to achieve a current density of 100 mA/cm^2^ and Tafel slope of 257 mV/dec. In HER, it also exhibited the highest catalytic activity with an overpotential of 42.1 mV at 10 mA/cm^2^ and Tafel slope of 61 mV/dec. Utilizing CuNi@Ni(ON)/CNTs-Gr electrodes in an alkaline electrolyzer requires only a low cell operating voltage of approximately 1.51 V to achieve a current density of 10 mA/cm^2^.

## 4. Conclusions and Perspectives

In conclusion, we introduced several strategies to improve the catalytic activity in water splitting, with a focus on the electrodeposition method, as summarized in Figure 10. This method has several advantages, including the ability to control the morphology, composition, and phases of the electrocatalyst, increase the number of active sites and surface area, directly deposit the electrocatalyst on the porous electrode, and facilitate large-scale synthesis. However, there are still several challenges that need to be addressed to make this method more convincing and versatile. One of the challenges is the unclear generation mechanism of nanostructures and heterointerfaces. To better understand the fundamental principles of electrodeposited catalysts, in situ and real-time monitoring techniques are needed. By combining advanced characterization techniques, such as in situ microscopy and operando X-ray analysis, with theoretical calculations, researchers can gain a deeper understanding of the relationship between materials and reaction mechanisms [155]. Another challenge is the need for the optimization of heterostructured electrocatalysts. Adopting new catalyst systems, such as high-entropy alloys, single-atom catalysts, and core–shell-like heterostructures can improve catalytic activity in water splitting due to their unique chemical and physical properties. However, to achieve industrialization, it is necessary to overcome the complex electrodeposition process parameters and establish a complete database of experimental parameters that can be applied to large-scale experiments [156]. For industrialization of the electrodeposition method, specialized equipment may be required for upscaling the electrodeposition process to an industrial scale. This may involve modifications to the existing infrastructure, such as the use of high-capacity power supplies and large-scale electrochemical reactors [34]. Developing a new, optimized catalyst system is one of the promising ways to utilize the electrochemical method for large-scale synthesis of water-splitting electrocatalysts. Lastly, the complexity of the synthesis process is a major challenge in finding the optimal conditions for electrodeposition. Machine learning has emerged as a promising solution to this problem, as it can help reduce time and material waste, as well as increase the chance of developing high-performance catalysts. By combining experimental results with machine learning, researchers can effectively design efficient and stable electrocatalysts [157].

We strongly believe that electrodeposition is the most promising way to prepare efficient water-splitting electrocatalysts for industrialization. Further theoretical and experimental studies combined with machine learning will likely open up new opportunities for the development of improved commercial water-splitting processes [122]. The challenges that need to be addressed, such as the unclear generation mechanism of nanostructures, the need for heterostructured electrocatalysts, and the complexity of the synthesis process, can be overcome using advanced techniques and cutting-edge technology.

## Figures and Tables

**Figure 1 materials-16-03044-f001:**
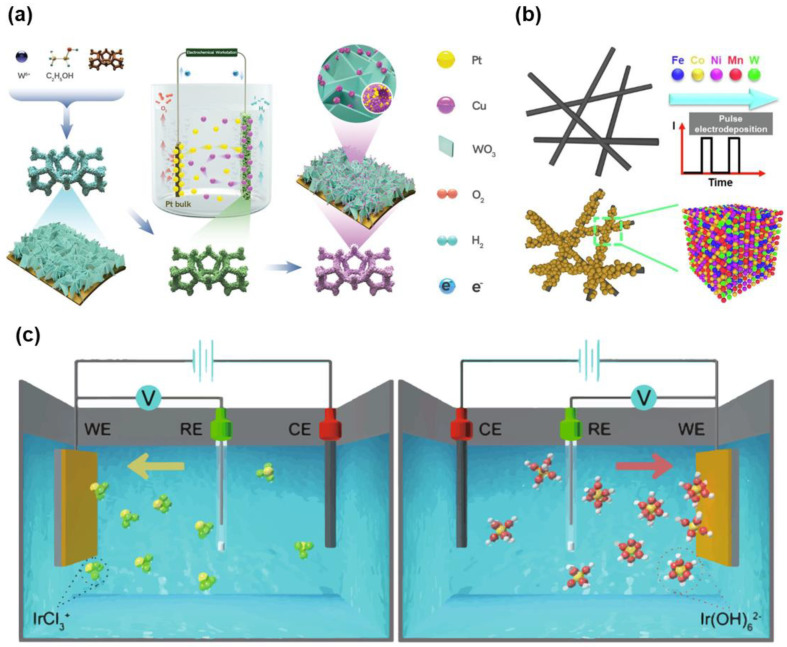
Schematic illustration of (**a**) the preparation of PtCu/WO_3_@CF (Pt loading: 0.0043 mg/cm^2^). Reprinted (adapted) from [109], copyright (2021) Wiley-VCH GmbH. (**b**) Fabrication of H-FeCo-NiMnW. Reprinted (adapted) from [110], copyright (2023) Elsevier B.V. (**c**) Electrochemical deposition of Ir species cathodic deposition and anodic deposition. Reprinted (adapted) from [111], copyright (2023) Springer Nature Limited.

**Figure 2 materials-16-03044-f002:**
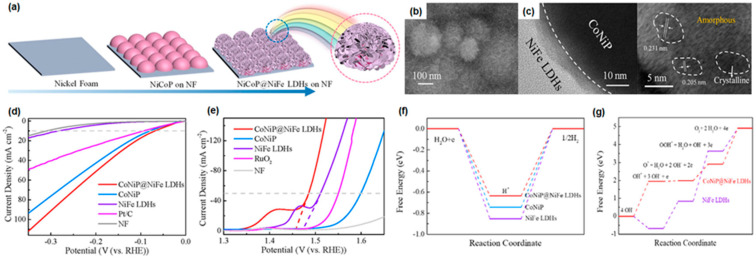
(**a**) The illustration for the preparation of CoNiP@NiFe LDH heterostructure directly on NF substrates through a two-step electrodeposition method. (**b**) Field-emission scanning electron microscope (FE-SEM) image of CoNiP@NiFe LDHs. (**c**) High-resolution transmission electron microscope (HR-TEM) image of CoNiP@NiFe LDHs. Polarization curves with 85% iR correction for (**d**) HER and (**e**) OER. (**f**) Gibbs free energy diagram vs. reaction coordinate of the HER with the c-CoNiP (001), c-NiFe LDHs (001), and c-CoNiP@NiFe LDHs. (**g**) Gibbs free energy diagram vs. reaction coordinate of the OER with the NiFe LDHs and c-CoNiP@NiFe LDHs. Reprinted (adapted) from [125], copyright (2023) Elsevier Ltd.

**Figure 3 materials-16-03044-f003:**
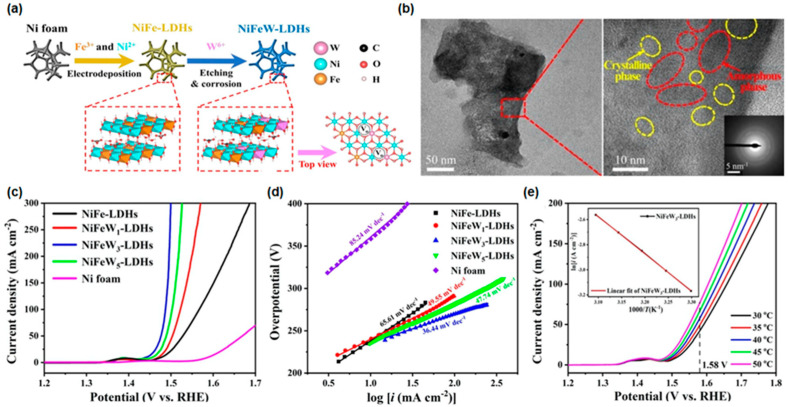
(**a**) Schematic diagram of a two-step synthesis process of NiFeW-LDHs electrocatalysts. (**b**) TEM images of NiFeW_3_-LDHs catalysts. (**c**) Polarization curves and (**d**) Tafel fitting curves after iR correction for the OER in alkaline medium. (**e**) Polarization curves at various temperatures and ln(current density) vs. 1/T fitting curves of NiFeW_3_-LDHs. Reprinted (adapted) from [127], copyright (2023) Elsevier B.V.

**Figure 4 materials-16-03044-f004:**
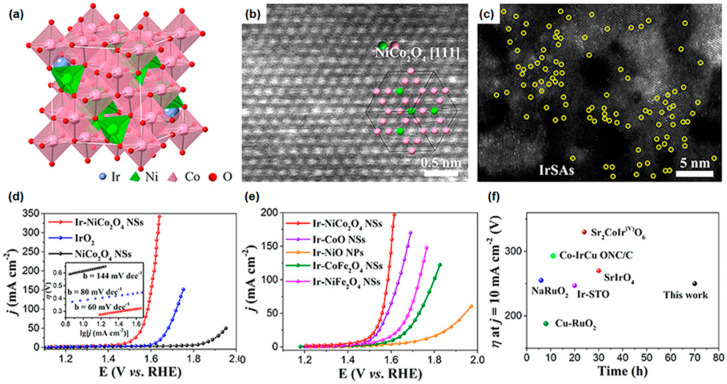
(**a**) Structural characterization of Ir-NiCo_2_O_4_ NSs. (**b**) High-angle annular dark-field scanning transmission electron microscope (HAADF-STEM) image of Ir-NiCo_2_O_4_ NSs and (**c**) HAADF-STEM image of Ir-NiCo_2_O_4_ NSs (Ir-SAs marked by yellow circles). (**d**) LSV curves of commercial IrO_2_, NiCo_2_O_4_, and Ir-NiCo_2_O_4_ NSs in acidic medium (0.5 M H_2_SO_4_). The inset in (**d**) shows the corresponding Tafel plots. (**e**) LSV curves of Ir SAs on different oxides substrate for the OER in 0.5 M H_2_SO_4_. (**f**) Compared overpotential at j = 10 mA/cm^2^ and stability of Ir-NiCo_2_O_4_ NSs with previously reported OER catalysts in acid media. Reprinted (adapted) from [134], copyright (2023) American Chemical Society.

**Figure 5 materials-16-03044-f005:**
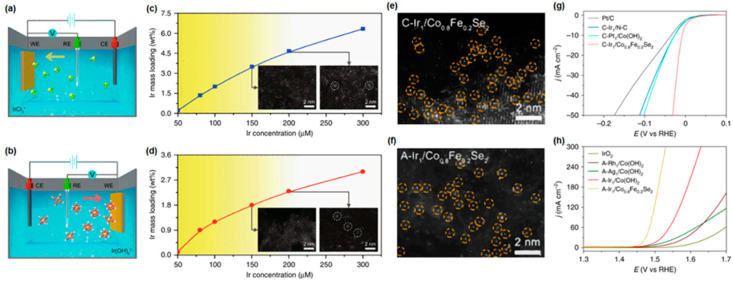
Schematic of (**a**) cathodic and (**b**) anodic deposition of Ir SACs. Differences in Ir mass loadings according to the concentration of IrCl_3_ for (**c**) cathodic and (**d**) anodic deposition. HR-TEM images of (**e**) cathodically and (**f**) anodically deposited Ir SAs on Co_0.8_Fe_0.2_Se_2_ support. Polarization curves of (**g**) cathodically deposited SACs for HER and (**h**) anodically deposited SACs for OER on various substrates. The Ir SAs on Co_0.8_Fe_0.2_Se_2_ support showed the best catalytic performance in both HER and OER (loading of cathodically and anodically deposited Ir SAs on Co_0.8_Fe_0.2_Se_2_: 2 wt.% and 1.2 wt.%, respectively). Reprinted (adapted) from [111], copyright (2023) Springer Nature Limited.

**Figure 6 materials-16-03044-f006:**
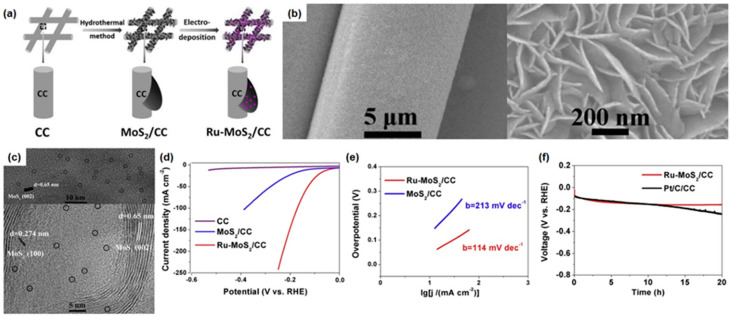
(**a**) The fabrication process of Ru-MoS_2_/carbon cloth array catalyst. (**b**) SEM images of Ru-MoS_2_/CC nanosheet. (**c**) HR-TEM images of Ru-MoS_2_/CC (single Ru atoms are marked by black circles). (**d**) Polarization HER curves of bare CC, MoS_2_/CC, and Ru-MoS_2_/CC (Ru loading: 46 μg/cm^2^) in 1.0 M KOH solution. (**e**) Tafel plots of MoS_2_/CC and Ru-MoS_2_/CC. (**f**) Chronopotentiometry analysis of long-term stability for Ru-MoS_2_/CC and Pt/C/CC at 10 mA/cm^2^ for 20 h. Reprinted (adapted) from [135], copyright (2023) Elsevier B.V.

**Figure 7 materials-16-03044-f007:**
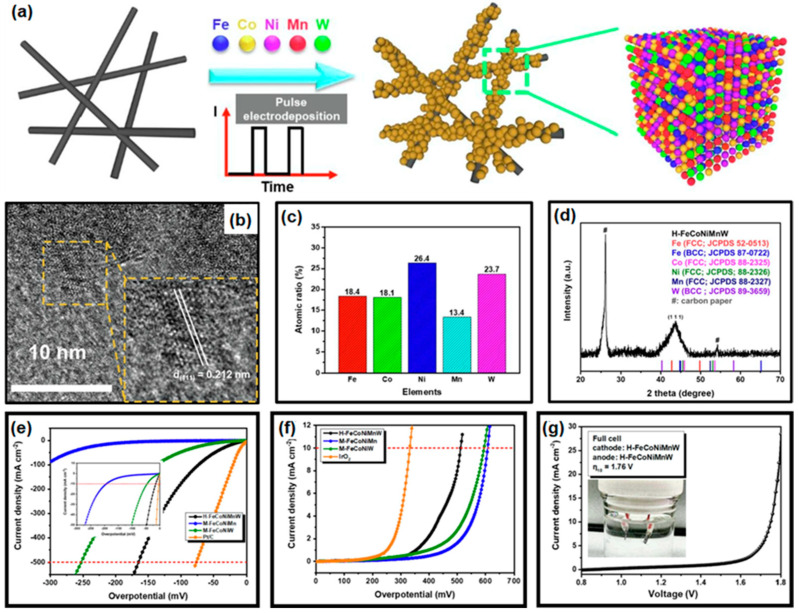
(**a**) Schematic illustration of the synthesis of H-FeCoNiMnW using pulsed mode electrodeposition. (**b**) HR-TEM image of electrodeposited H-FeCoNiMnW powder. (**c**) Atomic composition of H-FeCoNiMnW according to ICP-OES measurement. (**d**) XRD pattern of the synthesized H-FeCoNiMnW. (**e**) Electrocatalytic water-splitting performance of H-FeCoNiMnW, M-FeCoNiMn, and M-FeCoNiW in acidic medium (0.5 M H_2_SO_4_) (**e**) for HER and (**f**) for OER (H-FeCoNiMnW loading: 8 mg/cm^2^). (**g**) Overall water-splitting demonstration using H-FeCoNiMnW as both the cathode and the anode at 10 mA/cm^2^ (cell voltage: 1.76 V). Reprinted (adapted) from [110], copyright (2023) Elsevier B.V.

**Figure 8 materials-16-03044-f008:**
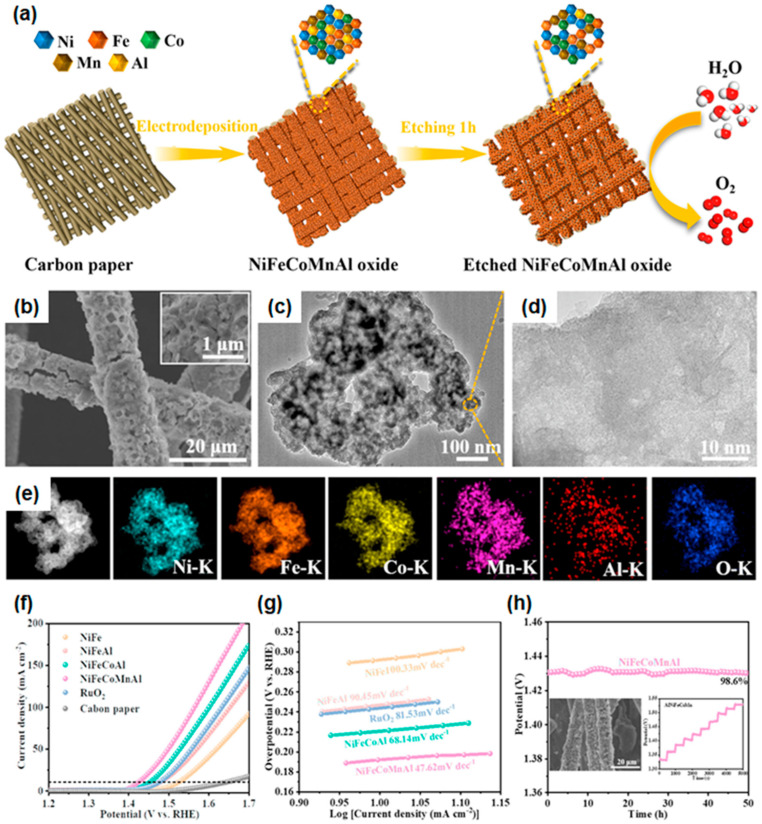
(**a**) Schematic illustration of the synthetic procedures for nanoporous NiFeCoMnAl oxide grown on carbon paper. (**b**) SEM and (**c**,**d**) TEM images of NiFeCoMnAl after soaking in 1.0 M KOH. (**e**) EDS elemental mapping of Ni, Fe, Co, Mn, Al, and O for etched NiFeCoMnAl. (**f**) LSV polarization curves and (**g**) Tafel plots of catalysts. (**h**) Stability of Ni-FeCoMnAl (NiFeCoMnAl loading: 0.35 mg/cm^2^). Reprinted (adapted) from [148], copyright (2023) Elsevier B.V.

**Figure 10 materials-16-03044-f010:**
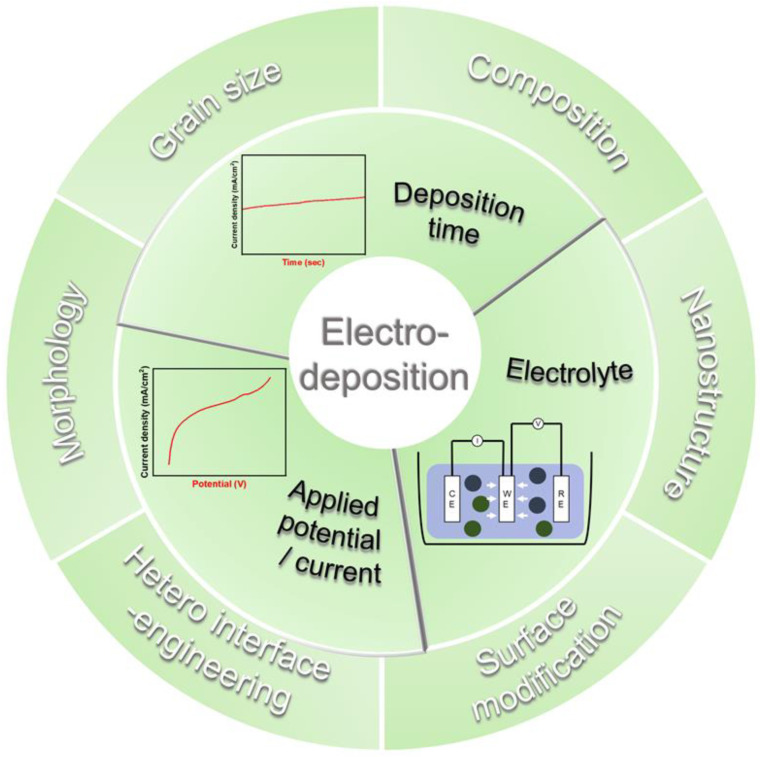
Summary of the various strategies and process variables in electrodeposition to improve water-splitting catalytic performance.

## Data Availability

Not applicable.

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
