# Peer review of "Recent Advances in Water-Splitting Electrocatalysts Based on Electrodeposition"

_materials, 2023, doi:10.3390/ma16083044_

Round 1

Reviewer 1 Report

The article "Recent Advances in Water-Splitting Electrocatalysts Based on Electrodeposition" is a comprehensive review of the latest developments in electrodeposited electrocatalysts for water-splitting. The authors provide an in-depth analysis of the various electrochemical methods used to produce high-performance electrocatalysts for the oxygen evolution reaction (OER) and hydrogen evolution reaction (HER). The review article also covers the principles of water electrolysis, the current status of electrocatalysts, and the challenges that need to be addressed for the development of efficient water-splitting technologies.

The authors have done an excellent job of presenting a detailed overview of the different electrochemical deposition techniques used for synthesizing electrocatalysts. They have also provided a comprehensive analysis of the various parameters involved in electrocatalyst synthesis, including the choice of precursor materials, deposition parameters, and post-treatment methods. The article also provides a critical evaluation of the different electrocatalysts reported in the literature, highlighting their strengths and weaknesses.

The review article is well-organized and written in a clear and concise manner. The authors have provided an extensive list of references, making it an excellent resource for researchers working in the field of electrocatalysis and water-splitting. The article also provides valuable insight into the challenges and limitations of electrocatalyst synthesis, which is particularly relevant for the development of efficient water-splitting technologies.

Overall, "Recent Advances in Water-Splitting Electrocatalysts Based on Electrodeposition" is a well-written and informative review article that provides a comprehensive analysis of the current state-of-the-art in electrocatalyst synthesis for water-splitting. The authors have presented a thorough and critical evaluation of the various electrocatalysts reported in the literature and have highlighted the challenges that need to be addressed for the development of efficient water-splitting technologies. The article is highly recommended for researchers and scientists working in the field of electrocatalysis and renewable energy.

Author Response

We appreciate the reviewer for taking the time to read our manuscript and the positive comment.

Reviewer 2 Report

In this article, authors review the mechanism and mode of electrodeposition, in addition, the synthesized various water-splitting electrocatalysts, including hydroxides, single-atom catalysts, high entropy alloys composite, by electrodeposition method are summarized. Currently, electrodeposition is a common method to construct electrocatalyst, this method can be helpful to design superior electrocatalyst. However, there still exist some issues authors should notice. I recommend this manuscript can be accepted after revision. The detailed comments are listed as following:

1. In the part of introduction, authors mention there are many current methods for the synthesis of transition metal-based catalysts, such as chemical vapor deposition, hydrothermal, corrosion, etc. Compared with these methods, the advantages of the electrodeposition method should be supplemented.

2. For water-splitting electrocatalysts, the water-splitting mechanism, including HER and OER, should be proposed.

3. Authors describe the application of electrodeposition method to prepare hydroxides, single-atom catalysts and high entropy alloys. Nevertheless, the correlation of electrodeposition with the structure-activity relationship of catalyst are not clear, which should be illustrated.

4. In the Figure captions, abbreviations of FE-SEM, HR-TEM and HAADF-STEM appear in the manuscript for the first time, the full name of these abbreviations should be supplied.

5. Some graphic elements associated with the contents should be added in the Figure 10.

Author Response

  1. In the part of introduction, authors mention there are many current methods for the synthesis of transition metal-based catalysts, such as chemical vapor deposition, hydrothermal, corrosion, etc. Compared with these methods, the advantages of the electrodeposition method should be supplemented.

We sincerely thank the reviewer for the insightful comment. We explained the advantages of electrodeposition compared to chemical vapor deposition, hydrothermal, corrosion, etc. Electrodeposition provides a range of advantages such as pricise control and versatility of the resulting materials.

Page 3, Line 117

à Compared to other methods, electrodeposition offers a range of advantages. Electrodeposition allows for precise control over the shape and size of the materials, making it ideal for obtaining the catalysts with specific dimensions and properties. Moreover, it can be utilized to synthesize a variety of materials including metals, oxides, sulfides, phosphides, and so on.

  1. For water-splitting electrocatalysts, the water-splitting mechanism, including HER and OER, should be proposed.

We appreciate the reviewer for the kind comment. Based on the reviewer's mentions, we perceived that it was essential to include the mechanisms of both HER and OER in the paper. As a result, we added the mechanisms of HER and OER in both acidic and alkaline media to the manuscript.

Page 2, Line 51

à Hydrogen evolution reaction (HER) and oxygen evolution reaction (OER) are two important surface reactions in water electrolysis. The mechanism of these reactions depends on the pH level of the electrolyte. The reaction of hydrogen and oxygen evolution in an acidic media follow below mechanism:

H+ + e→ Had

H+ + e + Had = H2

2Had → H2

H2O (l) + OH + H+ + e

OH → O + H+ + e

H2O (l) + O → OOH + H+ + e

OOH → + O2 (g) + H+ + e

where represents the active sites of the catalyst, (g) refers to the gas phase, and OH, O, and OOH represent the species adsorbed on the active site [33]. The HER mechanism involves the reduction of hydrogen ions (H+) to form hydrogen gas (H2) at the cathode. The OER mechanism, on the other hand, involves the oxidation of water (H2O) to form oxygen gas (O2) at the anode. The reaction of hydrogen and oxygen evolution in an alkaline media following mechanism:

H2O+e → OH+Had

H2O+e+Had → OH+H2

+ OHOH + e

OH + OHO + H2O(l) + e

O + OHOOH + e

OOH + OH + O2 (g) + H2O(l) + e

The HER mechanism involves the reduction of water (H2O) to form hydroxide ions (OH-) and hydrogen gas (H2) at the cathode. The OER mechanism involves the oxidation of hydroxide ions (OH-) to form oxygen gas (O2) and water (H2O) at the anode.

A following reference was added for supporting above paragraphs.

Page 22, Line 802

Reference 33 . Liang, Q.; Brocks, G.; Bieberle-Hütter, A. Oxygen Evolution Reaction (OER) Mechanism under Alkaline and Acidic Conditions. JPhys Energy 2021, 3, doi:10.1088/2515-7655/abdc85.

  1. Authors describe the application of electrodeposition method to prepare hydroxides, single-atom catalysts and high entropy alloys. Nevertheless, the correlation of electrodeposition with the structure-activity relationship of catalyst are not clear, which should be illustrated.

We sincerely thank the reviewer for the thoughtful comment. As mentioned by the reviewer, it is clear that we should have considered the impact of electrodeposition on the performance of the electrocatalyst, in addition to its advantages, when synthesizing it. After looking into several references, we have revised the manuscript.

Page 3, Line 124

à Interestingly, the unique structures and morphology such as nanowires, nanotubes, nanoparticles, and nanosheets achieved by electrodeposition enable to tailor the surface area, electronic properties, and surface reactivity, all of which affect the performance of catalysts. In addition, the atomic structure of electrodeposited catalysts can be elaborately modified to boost the activity of catalytic sites.

  1. In the Figure captions, abbreviations of FE-SEM, HR-TEM and HAADF-STEM appear in the manuscript for the first time, the full name of these abbreviations should be supplied.

We sincerely thank the reviewer for the thoughtful comment. As we initially used abbreviations without providing full names first, we have revised the manuscript to consider the convenience of readers. According to this comment, we have revised the manuscript and caption of Figure 2 and 4.

Page 6, Line 237

à (b) Field emission scanning electron microcope (FE-SEM) image of CoNiP@NiFe LDHs and (c) High-resolution transmission electon microscope (HR-TEM) image of CoNiP@NiFe LDHs. Polarization curves with 85% iR-correction for (d) HER and for (e) OER.

Page 7, Line 263

à High-resolution transmission electon microscope (HR-TEM) images showed that the amorphous NiFe LDH nanosheets well-covered the low-crystalline CoNiP nanoparticles and only a small amount of micro-crystalline region can be found, as displayed in Figure 2c.

Page 7, Line 300

à Scanning electron microscope (SEM) measurements showed changes in surface morphology, including the presence of nanopores, which can increase the contact area between the electrolytes and catalysts, resulting in faster desorption of OER products.

Page 9, Line 350

à (b) High-angle annular dark field- scanning transmission electron microscope (HAADF-STEM) image of Ir-NiCo2O4 NSs and (c) HAADF-STEM image of Ir–NiCo2O4 NSs (Ir-SAs marked by yellow circles).

  1. Some graphic elements associated with the contents should be added in the Figure 10.

We sincerely thank the reviewer for the insightful comment. Based on the comment, we have added some graphics to Figure 10 to make the conclusion of this review more visually comprehensible. According to this comment, we have revised the Figure 10.

Page 19, Line 707

Figure 10. Summary of the various strategies and process variables in electrodeposition to improve water-splitting catalytic performance.

Reviewer 3 Report

This review provides a pertinent introduction about recent progress on the field of electrodeposition for water splitting and offers solutions to current problems and the potential of electrodeposition in the water-splitting electrocatalysts. Considering the importance of electrodeposition in water-splitting, I would like to suggest this manuscript to be published after revision. Here are the detailed comments below:

1. The readers may want to know the significance of this review. Authors should clearly emphasize the significances of this review in the introduction section.

2. It would be better to replace Figure 1 because of the poor resolution and boxes on the picture.

3. Check the whole REFERENCES carefully. Some errors should be revised, such as the chemical formula in reference 3 and the lower index in reference 11.

4. It would be better to add an example other than LDHs in section 3.4, because LDHs have been focused on the section 3.1.

Author Response

  1. The readers may want to know the significance of this review. Authors should clearly emphasize the significances of this review in the introduction section.

We sincerely thank the reviewer for the insightful comment. For emphasizing the significances of this review, we introduced nanostructures and various techniques of electrodeposition discussed in this paper. According to this comment, we have revised the manuscript.

Page 3, Line 136

à Recently, layered double hydroxides, single atom catalysts, and high entropy alloys have attracted significant attention in the field of electrocatalysis due to their extraordinary catalytic activity derived from unique atomic structure [100]. Using various techniques in electrochemical deposition such as galvanostatic, potentiostatic, pulsed, and cyclic voltammetry modes, they can be uniformly synthesized and their catalytic activity can be boosted.

Page 27, Line 969

à Reference 100. Zhu, J.; Hu, L.; Zhao, P.; Lee, L.Y.S.; Wong, K.Y. Recent Advances in Electrocatalytic Hydrogen Evolution Using Nanoparticles. Chem. Rev. 2020, 120, 851–918, doi:10.1021/acs.chemrev.9b00248.

  1. It would be better to replace Figure 1 because of the poor resolution and boxes on the picture.

We appreciate the reviewer for your helpful comment. We tried to improve the resolution of Figure 1. Unfortunately, the original figure itselt shows the low quality of resolution.

  1. Check the whole REFERENCES carefully. Some errors should be revised, such as the chemical formula in reference 3 and the lower index in reference 11.

We thank the reviewer for the valuable comment. We checked the reference again and made corrections to the superscripts, subscripts, DOI, etc. According to this comment, we have revised the references.

Page 20, Line 728

  1. Seitz, L.C.; Dickens, C.F.; Nishio, K.; Hikita, Y.; Montoya, J.; Doyle, A.; Kirk, C.; Vojvodic, A.; Hwang, H.Y.; Norskov, J.K.; et al. A Highly Active and Stable IrOx/SrIrO3 Catalyst for the Oxygen Evolution Reaction. Science (80-. ). 2016, 353, 1011–1014.
  2. Voiry, D.; Fullon, R.; Yang, J.; De Carvalho Castro E Silva, C.; Kappera, R.; Bozkurt, I.; Kaplan, D.; Lagos, M.J.; Batson, P.E.; Gupta, G.; et al. The Role of Electronic Coupling between Substrate and 2D MoS2 Nanosheets in Electrocatalytic Production of Hydrogen. Nat. Mater. 2016, 15, 1003–1009, doi:10.1038/nmat4660.
  3. Han, A.; Zhou, X.; Wang, X.; Liu, S.; Xiong, Q.; Zhang, Q.; Gu, L.; Zhuang, Z.; Zhang, W.; Li, F.; et al. One-Step Synthesis of Single-Site Vanadium Substitution in 1T-WS2 Monolayers for Enhanced Hydrogen Evolution Catalysis. Nat. Commun. 2021, 12, 1–10, doi:10.1038/s41467-021-20951-9.
  4. Laursen, A.B.; Patraju, K.R.; Whitaker, M.J.; Retuerto, M.; Sarkar, T.; Yao, N.; Ramanujachary, K. V.; Greenblatt, M.; Dismukes, G.C. Nanocrystalline Ni5P4: A Hydrogen Evolution Electrocatalyst of Exceptional Efficiency in Both Alkaline and Acidic Media. Energy Environ. Sci. 2015, 8, 1027–1034, doi:10.1039/c4ee02940b.
  5. Shi, H.; Zhao, G. Water Oxidation on Spinel NiCo2O4 Nanoneedles Anode: Microstructures, Specific Surface Character, and the Enhanced Electrocatalytic Performance. J. Phys. Chem. C 2014, 118, 25939–25946, doi:10.1021/jp508977j.
  6. Jaramillo, T.F.; Jørgensen, K.P.; Bonde, J.; Nielsen, J.H.; Horch, S.; Chorkendorff, I.; Identification of Active Edge Sites for Electrochemical H2 Evolution from MoS2 Nanocatalysts. Science (80-. ). 2007, 317, 100 LP – 102.
  7. Armbrüster, M.; Kovnir, K.; Friedrich, M.; Teschner, D.; Wowsnick, G.; Hahne, M.; Gille, P.; Szentmiklósi, L.; Feuerbacher, M.; Heggen, M.; et al. Al13 Fe4 as a Low-Cost Alternative for Palladium in Heterogeneous Hydrogenation. Nat. Mater. 2012, 11, 690–693, doi:10.1038/nmat3347.
  8. Li, F.; Han, G.F.; Noh, H.J.; Lu, Y.; Xu, J.; Bu, Y.; Fu, Z.; Baek, J.B. Construction of Porous Mo3P/Mo Nanobelts as Catalysts for Efficient Water Splitting. Angew. Chemie - Int. Ed. 2018, 57, 14139–14143, doi:10.1002/anie.201808844.
  9. Yuan, Y.; Adimi, S.; Thomas, T.; Wang, J.; Guo, H.; Chen, J.; Attfield, J.P.; DiSalvo, F.J.; Yang, M. Co3Mo3N—An Efficient Multifunctional Electrocatalyst. Innov. 2021, 2, doi:10.1016/j.xinn.2021.100096.
  10. Shan, J.; Guo, C.; Zhu, Y.; Chen, S.; Song, L.; Jaroniec, M.; Zheng, Y.; Qiao, S.Z. Charge-Redistribution-Enhanced Nanocrystalline Ru@IrOx Electrocatalysts for Oxygen Evolution in Acidic Media. Chem 2019, 5, 445–459, doi:10.1016/j.chempr.2018.11.010.
  11. Meng, G.; Sun, W.; Mon, A.A.; Wu, X.; Xia, L.; Han, A.; Wang, Y.; Zhuang, Z.; Liu, J.; Wang, D.; et al. Strain Regulation to Optimize the Acidic Water Oxidation Performance of Atomic-Layer IrOx. Adv. Mater. 2019, 31, 1–8, doi:10.1002/adma.201903616.
  12. Wu, Y.; Liu, X.; Han, D.; Song, X.; Shi, L.; Song, Y.; Niu, S.; Xie, Y.; Cai, J.; Wu, S.; et al. Electron Density Modulation of NiCo2S4 Nanowires by Nitrogen Incorporation for Highly Efficient Hydrogen Evolution Catalysis. Nat. Commun. 2018, 9, doi:10.1038/s41467-018-03858-w.
  13. Xiao, Z.; Huang, Y.C.; Dong, C.L.; Xie, C.; Liu, Z.; Du, S.; Chen, W.; Yan, D.; Tao, L.; Shu, Z.; et al. Operando Identification of the Dynamic Behavior of Oxygen Vacancy-Rich Co3O4for Oxygen Evolution Reaction. J. Am. Chem. Soc. 2020, 142, 12087–12095, doi:10.1021/jacs.0c00257.
  14. Qi, K.; Cui, X.; Gu, L.; Yu, S.; Fan, X.; Luo, M.; Xu, S.; Li, N.; Zheng, L.; Zhang, Q.; et al. Single-Atom Cobalt Array Bound to Distorted 1T MoS2 with Ensemble Effect for Hydrogen Evolution Catalysis. Nat. Commun. 2019, 10, 1–9, doi:10.1038/s41467-019-12997-7.
  15. Du, H.; Zhang, X.; Tan, Q.; Kong, R.; Qu, F. A Cu3P-CoP Hybrid Nanowire Array: A Superior Electrocatalyst for Acidic Hydrogen Evolution Reactions. Chem. Commun. 2017, 53, 12012–12015, doi:10.1039/c7cc07802a.
  16. Mishra, I.K.; Zhou, H.; Sun, J.; Qin, F.; Dahal, K.; Bao, J.; Chen, S.; Ren, Z. Hierarchical CoP/Ni5P4/CoP Microsheet Arrays as a Robust PH-Universal Electrocatalyst for Efficient Hydrogen Generation. Energy Environ. Sci. 2018, 11, 2246–2252, doi:10.1039/c8ee01270a.
  17. Liang, X.; Zheng, B.; Chen, L.; Zhang, J.; Zhuang, Z.; Chen, B. MOF-Derived Formation of Ni2P-CoP Bimetallic Phosphides with Strong Interfacial Effect toward Electrocatalytic Water Splitting. ACS Appl. Mater. Interfaces 2017, 9, 23222–23229, doi:10.1021/acsami.7b06152.
  18. Zhou, P.; Zhang, Y.; Ye, B.; Qin, S.; Zhang, R.; Chen, T.; Xu, H.; Zheng, L.; Yang, Q. MoP/Co2P Hybrid Nanostructure Anchored on Carbon Fiber Paper as an Effective Electrocatalyst for Hydrogen Evolution. ChemCatChem 2019, 11, 6086–6091, doi:10.1002/cctc.201900948.
  19. Riyajuddin, S.; Azmi, K.; Pahuja, M.; Kumar, S.; Maruyama, T.; Bera, C.; Ghosh, K. Super-Hydrophilic Hierarchical Ni-Foam-Graphene-Carbon Nanotubes-Ni2P-CuP2 Nano-Architecture as Efficient Electrocatalyst for Overall Water Splitting. ACS Nano 2021, 15, 5586–5599, doi:10.1021/acsnano.1c00647.
  20. Rathore, D.; Sharma, M.D.; Sharma, A.; Basu, M.; Pande, S. Aggregates of Ni/Ni(OH)2/NiOOH Nanoworms on Carbon Cloth for Electrocatalytic Hydrogen Evolution. Langmuir 2020, 36, 14019–14030, doi:10.1021/acs.langmuir.0c02548.
  21. Zhang, D.; Li, H.; Riaz, A.; Sharma, A.; Liang, W.; Wang, Y.; Chen, H.; Vora, K.; Yan, D.; Su, Z.; et al. Unconventional Direct Synthesis of Ni3N/Ni with N-Vacancies for Efficient and Stable Hydrogen Evolution. Energy Environ. Sci. 2022, 15, 185–195, doi:10.1039/d1ee02013g.
  22. Wu, X.; Wang, Z.; Zhang, D.; Qin, Y.; Wang, M.; Han, Y.; Zhan, T.; Yang, B.; Li, S.; Lai, J.; et al. Solvent-Free Microwave Synthesis of Ultra-Small Ru-Mo2C@CNT with Strong Metal-Support Interaction for Industrial Hydrogen Evolution. Nat. Commun. 2021, 12, 1–10, doi:10.1038/s41467-021-24322-2.
  23. Oh, S.; Park, H.; Kim, H.; Park, Y.S.; Ha, M.G.; Jang, J.H.; Kim, S.K. Fabrication of Large Area Ag Gas Diffusion Electrode via Electrodeposition for Electrochemical CO2 Reduction. Coatings 2020, 10, 1–14, doi:10.3390/coatings10040341.
  24. Yan, J.; Kong, L.; Ji, Y.; White, J.; Li, Y.; Zhang, J.; An, P.; Liu, S.; Lee, S.T.; Ma, T. Single Atom Tungsten Doped Ultrathin α-Ni(OH)2 for Enhanced Electrocatalytic Water Oxidation. Nat. Commun. 2019, 10, 1–10, doi:10.1038/s41467-019-09845-z.
  25. He, L.G.; Cheng, P.Y.; Cheng, C.C.; Huang, C.L.; Hsieh, C.T.; Lu, S.Y. (NixFeyCo6-x-y)Mo6C Cuboids as Outstanding Bifunctional Electrocatalysts for Overall Water Splitting. Appl. Catal. B Environ. 2021, 290, doi:10.1016/j.apcatb.2021.120049.
  26. Murali, M.; Babu, S.P.K.; Krishna, B.J.; Vallimanalan, A. Synthesis and Characterization of AlCoCrCuFeZnx High-Entropy Alloy by Mechanical Alloying. Prog. Nat. Sci. Mater. Int. 2016, 26, 380–384, doi:10.1016/j.pnsc.2016.06.008.
  27. Zhou, D.; Cai, Z.; Jia, Y.; Xiong, X.; Xie, Q.; Wang, S.; Zhang, Y.; Liu, W.; Duan, H.; Sun, X. Activating Basal Plane in NiFe Layered Double Hydroxide by Mn2+ Doping for Efficient and Durable Oxygen Evolution Reaction. Nanoscale Horizons 2018, 3, 532–537, doi:10.1039/c8nh00121a.

  1. It would be better to add an example other than LDHs in section 3.4, because LDHs have been focused on the section 3.1.

We thank the reviewer for the valuable comment. We additionaly provided an example of core-shell structure synthesized by electrodeposition in section 3.4. According to this comment, we have revised the manuscript.

Page 17, Line 649

à Lee’s group have demonstrated CuNi@Ni(ON) core-shell heterostructures uniformly dispersing on 3D porous CNTs-Gr for alkaline HER and OER by electrodeposition method [155]. They conducted 8 CV cycles at the potential range of -0.5 to -1.0 V vs. Ag/AgCl with a scan rate of 5 mV/s in the solution of 1.0 mM Cu(Co2CH3)2 and 40 mM Ni(NO3)2 6H2O, resulting in the formation of CuNi@Ni core-shell nanoparticles. Then, a partial nitridation treatment were carried out in a quenching furnace at 400 ℃ for 2 h under the atmosphere of Ar (100 sccm) and NH3 (50 sccm). The resultant CuNi@Ni(ON) heterostructures on CNTs-Gr exhibited dual functionality in promoting both HER and OER. It is attributed to the alterations in the electronic structure of the surface, the type and number of electroactive sites, and charge conductivity. Moreover, the 3D porous and highly conductive CNTs-Gr substrate enhanced the stabilization of the active materials, facilitated hetero-charge transfer, and accelerated mass transfer. In OER, CuNi@Ni(ON)/CNTs-Gr showed the highest performance with an overpotential of 410 mV to achieve the current density of 100 mA/cm2 and the Tafel slope of 257 mV/dec. In HER, it also exhibited the highest catalytic activity with an overpotential of 42.1 mV at 10 mA/cm2 and the Tafel slope of 61 mV/dec. Utilizing CuNi@Ni(ON)/CNTs-Gr electrodes in an alkaline electrolyzer requires only a low cell operating voltage of approximately 1.51 V to achieve a current density of 10 mA/cm2 [155].

Page 31, Line 1104

à Reference 155. Tran, D.T.; Hoa, V.H.; Prabhakaran, S.; Kim, D.H.; Kim, N.H.; Lee, J.H. Activated CuNi@Ni Core@shell Structures via Oxygen and Nitrogen Dual Coordination Assembled on 3D CNTs-Graphene Hybrid for High-Performance Water Splitting. Appl. Catal. B Environ. 2021, 294, 120263, doi:10.1016/j.apcatb.2021.120263.

Reviewer 4 Report

Comments to authors

1. Line 269-270, “For the Tafel slope as illustrated in Figure 3d, the small value of 36.44 mV/dec can be calculated for NiFeW3 LDHs, indicating a good high-current electrocatalytic performance.” The authors should comment on what the obtained Tafel slope say about the mechanism involved in the hydrogen evolution on this study.

2. Line 277-278, “From the chronopotentiometry and cycle test for 120 h at 10 mA/cm2 and 5000 cycles, the NiFeW3-LDH retains their initial catalytic performance without significant changes.” The 1st cycle and the 5000th cycle should be compared to show the changes in performance during cycling.

3. In all the studies, the catalysts loading should be reported to enables readers to compare studies where there is comparable catalyst loading in relation to the performances of the catalyst’s materials.

4. In Figure 2 of section 3.1. Layered double hydroxides (LDHs), the images are not strongly visible. The authors should acquire high resolution images that would be much visible. Can the authors also comment on the agglomeration state of the LDH nanosheets and how this could affect the performance of the catalysts. The authors should also suggest a solution to reduce agglomeration as observed on the SEM image.

5. The authors should pay attention to the subscripts and superscripts on the manuscript. Some areas of the manuscripts lack these and should be fixed accordingly.

6. On the future perspectives, the authors should comment on the industrial developments towards using electrodeposition for catalysis formulation for hydrogen production. Where is industrial research at? What are the estimated costs when compared to other techniques used for hydrogen production? Is there specialized equipment that will be needed for upscaling to industrial scale?

Author Response

  1. Line 269-270, “For the Tafel slope as illustrated in Figure 3d, the small value of 36.44 mV/dec can be calculated for NiFeW3 LDHs, indicating a good high-current electrocatalytic performance.” The authors should comment on what the obtained Tafel slope say about the mechanism involved in the hydrogen evolution on this study.

We thank the reviewer for the valuable comment. As the reviewer mentioned, it is important to reveal the mechanism involved in the HER using Tafel slopes. However, as far as we know, it is difficult to reveal in OER related in Figure 3d. Instead, we have briefly added information about the reaction kinetics of tafel in oxygen evolution reaction and revised the manuscript accordingly. According to this comment, we have revised the manuscript.

Page 8, Line 318

à A lower Tafel slope value of NiFeW3 LDH compared to that of other samples indicates faster reaction kinetics in OER [130].

Page 29, Line 1040

à Reference 130.          Shinagawa, T.; Garcia-Esparza, A.T.; Takanabe, K. Insight on Tafel Slopes from a Microkinetic Analysis of Aqueous Electrocatalysis for Energy Conversion. Sci. Rep. 2015, 5, 1–21, doi:10.1038/srep13801.

  1. Line 277-278, “From the chronopotentiometry and cycle test for 120 h at 10 mA/cm2 and 5000 cycles, the NiFeW3-LDH retains their initial catalytic performance without significant changes.” The 1st cycle and the 5000th cycle should be compared to show the changes in performance during cycling.

We thank the reviewer for the valuable comment. We added a new figure to the reference to show the comparison between the 1st cycle and the 5000th cycle. According to this comment, we have revised the Figure 3.

Page 8, Line 332

Figure 3. (a) TEM images of NiFeW3-LDHs catalysts. (b) Polarization curves and (c) Tafel fitting curves after iR-correction for the OER in alkaline media. (d) Polarization curves at various temperatures and ln(current density) vs. 1/T fitting curves of NiFeW3-LDHs. (e) The polarization curves before and after 5000 CV cycles. Reprinted (adapted) from reference [121], copyright (2023) Elsevier B.V.

  1. In all the studies, the catalysts loading should be reported to enables readers to compare studies where there is comparable catalyst loading in relation to the performances of the catalyst’s materials.

We thank the reviewer for the valuable comment. To compare the effect of catalyst loading on performance more closely, we provided the exact values of catalyst loading in each caption. According to this comment, we have revised the caption of Figure 1, 4, 5, 6, 7, 8.

Page 5, Line 199

à Schematic illustration of (a) the preparation of PtCu/WO3@CF (Pt loading: 0.0043 mg/cm2).

Page 9, Line 350

à (d) LSV curves of commercial IrO2, NiCo2O4, and Ir–NiCo2O4 NSs in acidic media (0.5 M H2SO4). The inset in Fig. 4d shows the corresponding Tafel plots. (Ir loading: 0.0104 mg/cm2 in total loding: 2.5104 mg/cm2).

Page 10, Line 395

à Polarization curves of (g) cathodically deposited SACs for HER and (h) anodically depos-ited SACs for OER on various substrates. The Ir SAs on Co0.8Fe0.2Se2 support shows the best catalytic performance in both HER and OER. (Cathodically and anodically deposited Ir SAs on Co0.8Fe0.2Se2 loading: 2 wt% and 1.2 wt%, respectively).

Page 11, Line 437

à (d) Polarization HER curves of bare CC, MoS2/CC (MoS2 loading: 12.4 mg/cm2), and Ru-MoS2/CC (Ru loading: 46 μg/cm2) in 1.0 M KOH solution.

Page 12, Line 480

à (e) Electrocatalytic water splitting performances of H-FeCoNiMnW, M− FeCoNiMn, and M− FeCoNiW in acidic media (0.5 M H2SO4) (e) for HER and (f) for OER (H-FeCoNiMnW loading: 8 mg/cm2).

Page 14, Line 537

à (f) LSV polarization curves and (g) the Tafel plots of catalysts and (h) The stability of Ni-FeCoMnAl (NiFeCoMnAl loading: 0.35 mg/cm2).

  1. In Figure 2 of section 3.1. Layered double hydroxides (LDHs), the images are not strongly visible. The authors should acquire high resolution images that would be much visible. Can the authors also comment on the agglomeration state of the LDH nanosheets and how this could affect the performance of the catalysts. The authors should also suggest a solution to reduce agglomeration as observed on the SEM image.

We thank the reviewer for the valuable comment. We tried to improve the resolution of Figure 2. Unfortunately, the original figure itselt shows the low quality of resolution. As the reviewer mentioned, we commented on the agglomeration state of the LDH nanosheets and its effect on the performance of the catalysts. In addition, we suggested a solution to suppress the agglomeration. According to this comment, we have revised the manuscript.

Page 6, Line 257

à Figure 2b shows the vertically aligned NiFe LDH nanosheets grown on the surface of elec-trodeposited CoNiP nanoparticles which are stacked on the nickel foam. In this image, the agglomeration of the LDH nanosheets is observed. The LDH nanosheets tend to agglomerate due to strong inter-layer van der Waals forces, resulting in decreased surface area and performance. Therefore, it is required to obtain uniformly dispersed LDH nanosheets using surfactants and pH control agents [126].

Page 29, Line 1029

à Reference 126. Wang, B.; Zhang, H.; Evans, D.G.; Duan, X. Surface Modification of Layered Double Hydroxides and Incorporation of Hydrophobic Organic Compounds. Mater. Chem. Phys. 2005, 92, 190–196, doi:10.1016/j.matchemphys.2005.01.013.

  1. The authors should pay attention to the subscripts and superscripts on the manuscript. Some areas of the manuscripts lack these and should be fixed accordingly.

We thank the reviewer for the valuable comment. Following the reviewer's comment, we carefully checked and revised all the citations, references, and figures to ensure their accuracy for the readership.

  1. On the future perspectives, the authors should comment on the industrial developments towards using electrodeposition for catalysis formulation for hydrogen production. Where is industrial research at? What are the estimated costs when compared to other techniques used for hydrogen production? Is there specialized equipment that will be needed for upscaling to industrial scale?

We thank the reviewer for the valuable comment. Based on our current research progress, we cannot provide specific cost values for comparing electrodeposition with other synthesis methods. However, we concluded that the electrodeposition has the potential for industrial applications and explained the reasons for this in the introduction. In addition, we also discussed some equipment that could be a solution for mass production, which is one of the limitations of industrializing electrodeposition in the conclusion. According to this comment, we have revised the manuscript.

Page 3, Line 128

à To further elaborate, electrodeposition is being explored as a method to increase produc-tion for industrial applications. Recently, a research paper reported the successful electro-deposition of an electrocatalyst onto a substrate that was 136 cm2 in size [97]. This is a no-ticeable figure even in the previous research, and based on this, nanostructure design is actively used in the research field of electrocatalysts to obtain excellent performance.

Also, the use of electrodeposition for water-splitting catalysts can be a promising candidate for the development of cost-effective and efficient water-splitting catalysts for industrial-scale hydrogen product [98-99].

Page 18, Line 687

à For industrialization of electrodeposition method, specialized equipment may be required for upscaling the electrodeposition process to an industrial scale. This may involve modifications to the existing infrastructure, such as the use of high-capacity pow-er supplies and large-scale electrochemical reactors [34].

Page 22, Line 804

à Reference 34.  She, Z.W.; Kibsgaard, J.; Dickens, C.F.; Chorkendorff, I.; Nørskov, J.K.; Jaramillo, T.F. Combining Theory and Experiment in Electrocatalysis: Insights into Materials Design. Science (80-. ). 2017, 355, doi:10.1126/science.aad4998.

Page 27, Line 962

à Reference 97. Oh, S.; Park, H.; Kim, H.; Park, Y.S.; Ha, M.G.; Jang, J.H.; Kim, S.K. Fabrication of Large Area Ag Gas Diffusion Electrode via Electrodeposition for Electrochemical CO2 Reduction. Coatings 2020, 10, 1–14, doi:10.3390/coatings10040341.

Reference 98. Shi, Y.; Huang, W.M.; Li, J.; Zhou, Y.; Li, Z.Q.; Yin, Y.C.; Xia, X.H. Site-Specific Electrodeposition Enables Self-Terminating Growth of Atomically Dispersed Metal Catalysts. Nat. Commun. 2020, 11, 1–9, doi:10.1038/s41467-020-18430-8.

Reference 99. Peters, B.K.; Rodriguez, K.X.; Reisberg, S.H.; Beil, S.B.; Hickey, D.P.; Kawamata, Y.; Collins, M.; Starr, J.; Chen, L.; Udyavara, S.; et al. Scalable and Safe Synthetic Organic Electroreduction Inspired by Li-Ion Battery Chemistry. Science (80-. ). 2019, 363, 838–845, doi:10.1126/science.aav5606.

Round 2

Reviewer 3 Report

After the revision, the manuscript has been much improved, and now it can be accepted.